# Exploring the Risk: Investigating the Association Between Elderly-Onset Sarcoidosis (EOS) and Malignancy

**DOI:** 10.3390/arm94010003

**Published:** 2026-01-02

**Authors:** Ahmed Ehab, Axel T. Kempa, Ahmad Shalabi, Noha Elkateb, Nesrine Saad Farrag, Heba Wagih Abdelwahab

**Affiliations:** 1Pulmonary Medicine Department, SLK Lung Center Loewenstein, 74245 Loewenstein, Germany; axel.kempa@rbk.de; 2Pulmonary Medicine Department, Mansoura University, Mansoura 35516, Egypt; 3DGD Lung Center Hemer, 58675 Hemer, Germany; 4Pulmonary Medicine Department, Robert Bosch Hospital, 70173 Stuttgart, Germany; 5Department of Thoracic Surgery, Mainz University Hospital, Johannes Gutenberg University Mainz, 55116 Mainz, Germany; ahmad.shalabi@unimedizin-mainz.de; 6Clinic for Internal Medicine, Rheumatology and Immunology, Medius Klinik, 73230 Kirchheim unter Teck, Germany; 7Community Medicine Department, Port Said University, Port Said 31111, Egypt

**Keywords:** sarcoidosis, younger-onset sarcoidosis, elderly-onset sarcoidosis, malignancy, risk factors

## Abstract

**Highlights:**

**What are the main findings?**
Patients with EOS were more frequently female and exhibited a markedly higher prevalence of prior malignancy (OR ≈8.0) compared to younger-onset counterparts. First bullet.Increasing age at sarcoidosis diagnosis was independent predictor of prior malignancy history, whereas sex, smoking history, and cardiometabolic profile did not significantly influence the likelihood of pre-existing neoplasia.

**What is the implication of the main finding?**
In elderly patients with newly diagnosed sarcoidosis—particularly those with a history of malignancy—histological confirmation is crucial to distinguish sarcoidosis from tumor recurrence or metastasis due to overlapping radiological findings.The increased prevalence of malignancy in elderly-onset sarcoidosis appears predominantly age-related rather than EOS-specific, emphasizing the need for age-adjusted risk assessment and cautious consideration of targeted cancer evaluation in future prospective studies.

**Abstract:**

**Background:** Elderly-onset sarcoidosis > 65 (EOS) is rare and occurs in patients over 65. Studies on its incidence, clinical features, and treatment are limited, and its link to malignancy remains complex. **Objectives:** In this study, we aimed to analyze the possible association between malignancy and the occurrence of sarcoidosis in elderly patients over 65 years old. **Design:** Monocentric, nested retrospective case–control study. **Material and Methods**: A retrospective study analyzed newly diagnosed sarcoidosis patients in the Loewenstein Lung Center, Baden-Württemberg, Germany, categorizing them into younger-onset (<65 years) and elderly-onset (≥65 years). Demographic data, smoking status, medical history, symptoms, diagnostic methods, and any prior malignancy history were collected. **Results:** A total of 447 patients were included (365 patients within the group of younger-onset sarcoidosis and 82 patients with EOS). The median age of the younger-onset group was 47 (47 [23–63] years), compared to 69 (69 [65–84] years), *p* ≤ 0.001. Female patients were more prevalent in the group of elderly-onsets (54.9%) compared to the younger-onset group (35.9%), corresponding to an odds ratio of 2.2 (95% CI: 1.3–3.5, *p*: 0.002). Regarding the past history of malignancy, patients who had a positive history of malignancy were more prevalent among the elderly-onset group (29.6%) compared to the younger-onset group (5%) [OR (95% CI): 8.1 (4.1–15.8), *p* ≤ 0.001]. In multivariable logistic regression analysis with malignancy as the outcome, increasing age at sarcoidosis diagnosis was independently associated with a higher likelihood of prior malignancy (adjusted OR 1.08 per year, 95% CI 1.04–1.12), whereas sex, smoking status, and cardiometabolic comorbidity (diabetes and/or hypertension) were not independently associated. **Conclusions:** Elderly-onset sarcoidosis (EOS) is a less frequent variant of sarcoidosis with limited data regarding the possible risk factors. The increased prevalence of malignancy observed among patients with elderly-onset sarcoidosis appeared to be largely driven by age rather than a distinct EOS-specific effect. Age-adjusted analyses are essential when interpreting malignancy risk in sarcoidosis, and future age-matched prospective studies are needed to clarify potential biological links and guide evidence-based screening strategies.

## 1. Introduction

Sarcoidosis is a multisystem inflammatory disorder of unknown etiology. The diagnosis is typically based on compatible clinical and radiological features. However, a definitive diagnosis is established through histological identification of non-caseating epithelioid cell granulomas, which serve as the pathological hallmark of sarcoidosis [1].

Sarcoidosis is a pervasive disease characterized by significant variability in the clinical presentation, ranging from asymptomatic cases to severe dysfunction of the affected organ(s). The condition arises from an unpredictable immune response, wherein unknown putative antigens in a genetically predisposed host trigger a T-helper 1 (Th1) cellular immune response, leading to the formation of characteristic granulomas [2]. The interplay between genetic and environmental triggers influences the risk of the disease, the spectrum of clinical symptoms, as well as clinical course and prognosis [3]. The estimated prevalence of sarcoidosis ranges from 50 to 160 per 100,000 individuals. The variability in prevalence is affected by geographical location, ethnicity, genetic factors, sex, and age demographics [4,5]. Numerous studies have reported that the peak age for the onset of sarcoidosis is between 30 and 40 years old, regardless of geographical distribution [6,7,8,9]. A second incidence peak of the disease, predominantly described in females, occurs between 50 and 65 years old, as reported in two studies [10,11].

Therefore, sarcoidosis in elderly patients is less common. The term elderly-onset sarcoidosis (EOS) is used to define the first diagnosis of sarcoidosis in patients over 65 years old, which is the most commonly used cut-off age for this definition [12,13]. The data on the incidence and prevalence of EOS remains limited and somewhat contradictory. In one study, a prevalence of EOS was estimated to be about 7% after retrospective analysis of 668 patients over 42 years [14]. EOS is likely underdiagnosed as it represents a clinical subset of patients with a distinct clinical presentation. The aging process may alter both clinical and radiological findings, and the common coexistence of comorbidities further complicates the diagnosis [12,15,16]. Unlike sarcoidosis in younger patients, histological confirmation of sarcoidosis is crucial in elderly patients for not only the establishment of the diagnosis but also for the exclusion of malignancy [12].

The potential association between sarcoidosis and malignancy remains controversial. A significant association between sarcoidosis and malignancy, particularly skin and hematologic neoplasm, was quantified in a meta-analysis, which analyzed 25,000 patients in 16 original studies [3]. Cohort studies showed that sarcoidosis patients were at a higher risk of cancer, especially lymphoma, in comparison to the general population [17,18,19]. A possible explanation of the relationship between sarcoidosis and malignancy is the presence of chronic inflammation and immune dysfunction in genetically susceptible patients [3]. The chronic inflammatory response promotes a carcinogenic role with stimulation of angiogenesis [3,20]. On the other hand, the development of sarcoidosis or a sarcoid-like reaction in cancer patients in the draining lymph nodes of the primary tumor or in distant sites was reported. Additionally, the coexistence of granuloma reaction in the primary tumor was also described [18,21].

To our knowledge, the association between the malignancy and EOS has not been investigated. Therefore, the aim of this study is to analyze the possible association between malignancy and the occurrence of sarcoidosis in elderly patients over 65 years old presented in our institute (in the Loewenstein lung center, Löwenstein, Baden-Württemberg, Germany) in the last 10 years.

## 2. Patients and Methods

We conducted a monocentric, nested retrospective case–control study to evaluate the potential association between the history of malignancy and EOS in patients aged over 65 years old.

Ethics approval and consent to participate: This study was approved by the Ethics Committee of the State Medical Association of Baden-Württemberg (F-2024-061, Stuttgart, Germany), and all methods were performed in accordance with the relevant guidelines and regulations. Informed consent to participate was waived by the ethics committee due to the retrospective nature of the study. The reporting of this study conforms to the Strengthening the Reporting of Observational Studies in Epidemiology (STROBE) statement.

### 2.1. Patients

A list of 714 patients was obtained after searching the institutional database in the Loewenstein Lung Center, Baden-Württemberg, Germany, using the diagnosis code for sarcoidosis (ICD-10-Code: D86.0) with hospital admission between 2013 and 2023. After analysis of the patients, the following patients were excluded: 178 patients with a previous history of sarcoidosis; in 89 patients, the sarcoidosis was not histologically confirmed. Therefore, a total number of 447 patients were finally included, fulfilling the following criteria: first presentation and histologically confirmed sarcoidosis diagnosis by the presence of non-caseating granuloma along with clinical and radiological features consistent with sarcoidosis after exclusion of other possible causes of granulomatous lung diseases (Figure 1).

The included patients were categorized into two groups based on their age at the time of their first presentation:Control group (younger-onset sarcoidosis): patients younger than 65 years old.Case group (elderly-onset sarcoidosis): patients 65 years old and older (EOS).

Data collection:

We gathered the following data from the medical records of the included patients:Demographic and relevant clinical data: age, sex, smoking status, hypertension (HTN), diabetes mellitus (DM), medical histories other than cancer, diffusion capacity of carbon monoxide (DLCO), forced expiratory volume in the first second (FEV1), and forced vital capacity (FVC).Sarcoidosis: leading symptoms, location of the enlarged lymph nodes, radiological stage of the pulmonary sarcoidosis, diagnostic method of the sarcoidosis, and finally initiation of the treatment of the sarcoidosis after multidisciplinary discussion (MDD).History of malignant tumors before the sarcoidosis diagnosis, when positive: type of malignancy, staging, treatment (surgery, chemotherapy, radiotherapy), and average time between both malignancy and development of sarcoidosis.

The levels of angiotensin-converting enzyme (ACE), neopterin, and serum interleukin 2 receptor (sIL-2r) were planned to be collected; however, the data were missing in many patients and could not be finally incorporated into the study.

### 2.2. Statistical Analysis

Data was analyzed using SPSS v.26. Categorical data was presented in the form of frequencies and percentages. Normality of continuous data was checked using the Shapiro–Wilk test, and normally distributed data was presented as mean (SD), while non-normally distributed data was presented as median (min–max). Whenever appropriate, significance testing was performed either using the chi-squared test or Fisher’s exact test for categorical data. While Welch’s t-test or the Mann–Whitney U test was used for significance testing in continuous data. Malignancy types were analyzed descriptively and stratified by sex. Sex-specific tumors (breast, ovarian, and prostate cancers) were analyzed within the relevant sex strata, while non-sex-specific tumors were analyzed in the overall cohort. Each tumor type was compared with the absence of malignancy within the same stratum. Due to small numbers and zero-cell counts, Fisher’s exact test was used, and odds ratios with 95% confidence intervals were calculated using the Haldane–Anscombe continuity correction. These analyses were considered exploratory. To assess the independent association between malignancy history and patient characteristics, multivariable logistic regression analyses were performed with prior malignancy as the dependent variable. Age at sarcoidosis diagnosis was modeled as a continuous variable to appropriately account for its confounding effect. Sex, smoking status, and comorbidity burden were included as covariates based on clinical relevance and prior literature. Given the limited number of malignancy events, the comorbidity burden was summarized using a composite cardiometabolic variable (diabetes mellitus and/or hypertension) to avoid model overfitting. Adjusted odds ratios (ORs) with 95% confidence intervals (CIs) were reported. As a sensitivity analysis, an alternative model using the age-based EOS classification (≥65 vs. <65 years) was conducted to examine the robustness of the findings. Statistical significance was defined as a two-sided *p* value < 0.05.

## 3. Results

The study included 365 patients within the group of younger-onset sarcoidosis and 82 patients with elderly-onset sarcoidosis. The median age of the younger-onset group was 47 (47 [23–63] years), compared to 69 (69 [65–84] years), *p* ≤ 0.001. Female patients were more prevalent in the group of elderly-onset (54.9%) compared to the younger-onset group (35.9%), corresponding to an odds ratio of 2.2 (95% CI: 1.3–3.5, *p* = 0.002). Ex-smokers and hypertensive and diabetic patients were more prevalent in the elderly-onset sarcoidosis group (49.1%, 75.6%, and 30.3%, respectively) compared to the other group (23.7%, 19.8, and 8.3%, respectively) [crude OR (95% CI): 2.4 (1.3–4.5), 12.6 (7.0–22.4), 4.8 (2.6–8.9), *p* = 0.005, ≤0.001, ≤0.001, respectively]. There were significant differences between both groups regarding the history of some other comorbidities, such as coronary artery disease (CAD) and chronic obstructive pulmonary disease (COPD) [crude OR (95% CI): 6.5 (2.2–19.2) and 4.724 (1.483–15.043); *p* = 0.001, 0.009, respectively]. There were no significant differences regarding having hypothyroidism, OSA, or BA (*p* = 0.075, 0.22, and 0.106, respectively) (Table 1).

The differences in clinical presentation between the two groups were demonstrated in Table 2. Regarding the leading presenting symptoms, there were no significant differences between the two groups regarding most symptoms. However, dyspnea was more common in elderly-onset (50.6%) compared to 19.4% in younger-onset (*p* ≤ 0.001), while Loefgren Syndrome was more common in younger-onset (13.6%) compared to elderly-onset (2.5%), *p* (0.005). Enlarged lymph nodes and parenchymal infiltration were more common in younger-onset, but with no significant difference (0.070, 0.408, respectively). Regarding the staging of sarcoidosis, presentation at later stages III and IV was more common within the elderly-onset group (12.3% and 6.4%, respectively) compared to the younger-onset group (5.2%, 1.4%) (*p* ≤ 0.001). Regarding lung functions, there were no significant differences except for FVC (L) and FEV1 (L). FVC (L), which was higher among the younger-onset group [median (min–max) 3.8 (0.37–89)] compared to the elderly-onset group [median (min–max) 2.7 (0.9–4.9)], *p* ≤ 0.001. Also, FEV1 (L), which was higher among the younger-onset group [mean (SD): 3.19 (0.93)] compared to the elderly-onset group [mean (SD): 2.27 (0.85)], *p* ≤ 0.001

Regarding the diagnostic methods used (Table 3), results show that endobronchial ultrasonography-guided transbronchial needle aspiration (EBUS-TBNA), endobronchial biopsy (EBB), and bronchoalveolar lavage (BAL) were significantly higher in the group of younger-onset compared to the group of elderly-onset sarcoidosis (*p* = 0.037, 0.010, ≤0.001), while mediastinoscopy, video-assisted thoracoscopic surgery (VATS)/open lung biopsy, CT-guided biopsy, and lymph node (LN) resection were higher in the group of elderly-onset sarcoidosis compared to the younger-onset group, but with significant difference only for open lung biopsy (*p* = 0.041).

The association between the onset of sarcoidosis and the history of malignancy is shown in Table 4. Patients who had a positive history of malignancy were more prevalent among the elderly-onset group (29.6%) compared to the younger-onset group (5%) [crude OR (95% CI): 8.1 (4.1–15.8), *p* ≤ 0.001]. There was a significant difference between both groups regarding the number of malignancies, where patients with one malignancy represented 5% of the younger-onset group compared to 21% in the elderly-onset group (*p* ≤ 0.001), and no patients had multiple malignancies in the younger-onset group, while in the elderly-onset group, multiple malignancies were present in 8.3% of the group. With the elderly-onset sarcoidosis, receiving chemotherapy (5%), operative resection (23.3%), radiotherapy (5%), or adjuvant immunotherapy (2.5%) was significantly higher compared to receiving chemotherapy (1.7%), operative treatment (3.9%), radiotherapy (1.1%), or adjuvant immunotherapy (0%) in younger-onset sarcoidosis (*p* = 0.035, ≤0.001, 0.013, 0.021). The duration of malignancy before sarcoidosis diagnosis was longer in elderly-onset sarcoidosis (median: 4.17 Y) compared to younger-onset sarcoidosis (median: 3.3 Y), with no significant difference (*p* = 0.907).

The distribution of malignancy in both groups is illustrated in Table 5 and Figure 2. The most common malignancy of all types was breast cancer, being 7.4% of the elderly-onset group and 1.1% in the younger-onset group, with significant differences between both groups (*p* = 0.013). Also, significant differences were observed among both groups regarding prostate cancer among the elderly-onset-only group, representing 11.1% of males in the elderly-onset group, with no cases of prostate cancer in the younger-onset group (*p* = 0.001). Renal cell carcinoma was present in 3.7% and 0.3% of the elderly and younger-onset groups, with a significant difference (*p* = 0.011). Otherwise, no significant differences were observed. Given the small number of events, these findings are presented as exploratory.

Table 6 shows the differences between the patients with positive and negative histories of malignancy within each of the groups of the study (younger and elderly-onset). Results show that cases with a positive history of malignancy were of higher age than those with a negative history, with a significant difference only for the group of younger-onsets (*p*: 0.010). Females were more represented in the group with positive history (61.1%) compared to 34.5% within the group of younger-onset, but no significant difference was found in the group of elderly-onset. There were no significant differences between the groups of younger and elderly-onset regarding smoking and comorbidities, apart from CAD within the group of elderly-onset, which was higher (20.8%) among cases with a positive history of malignancy compared to (5.3%) among patients with a negative history of malignancy (*p* = 0.046). There were no significant differences in the presentation of the leading symptoms within the group of elderly-onset. However, dyspnea and cough were significantly higher among the cases with a negative history of malignancy within the group of younger-onset (*p* = 0.030, 0.016, respectively). Results also show that regarding the diagnostic methods, no significant differences were found between the patients with positive and negative histories of malignancy within each of the groups of the study (younger and elderly-onset). Regarding the CT presentation, there were no significant differences between the two groups regarding enlarged LN and staging. However, parenchymal infiltration was higher among cases with no malignancy compared to cases with malignancy within both groups of younger and elderly-onset (*p* = 0.014, 0.034). The results show no significant differences regarding lung functions within both groups.

Table 7 showed in multivariable logistic regression analysis adjusting for sex, smoking status, and cardiometabolic comorbidity, increasing age at sarcoidosis diagnosis was independently associated with a higher likelihood of malignancy preceding sarcoidosis (adjusted OR 1.08 per year, 95% CI 1.04–1.12, *p* < 0.001). Sex, smoking status, and diabetes and/or hypertension were not independently associated with prior malignancy. In a sensitivity analysis using the age-based EOS classification, EOS was associated with higher odds of prior malignancy; however, this association was attenuated after accounting for age as a continuous variable.

Outcome variable: malignancy diagnosed before sarcoidosis. Age modeled as a continuous variable. Cardiometabolic comorbidity is defined as diabetes mellitus and/or hypertension.

## 4. Discussion

The relationship between sarcoidosis and malignancy is indeed complex and multifaceted, with the precise nature of this association remaining ambiguous. It is challenging to distinguish whether the observed co-occurrence is due to causality or mere coincidence [18]. Sarcoidosis, along with sarcoid-like reaction and other granulomatous inflammation, can not only mimic malignancies but has also been observed to develop in the regional lymph nodes as well as distant sites before, during, and after the onset of cancerous lesions [18]. Additionally, the phenomenon known as sarcoidosis-lymphoma syndrome, first described by Brinkcker, exemplifies this intricate correlation [22]. This phenomenon adds further complexity to diagnosis and management, necessitating careful clinical evaluation to distinguish between sarcoidosis and malignancy, especially in cases where both conditions may coexist.

The novel point in our work is the aim to evaluate the association between the malignancy and EOS, which is considered a rare subtype of sarcoidosis and was estimated to be in 7–30% of all sarcoidosis patients [12,14]. We found that patients with a history of malignancy were more prevalent among the EOS group (29.6%) compared to the younger-onset group (5%). In addition, breast cancer was present in 15.4% of the EOS and 3.3% in the younger-onset group, while ovarian cancer was present in 5.7% in the EOS group. In agreement with our results, breast cancer and lymphoma were the most frequently detected malignancies in a series of 29 sarcoidosis cases with pre-existing cancer [23]. Tumor-specific findings are descriptive and exploratory. Given the limited number of events, these results should not be interpreted as evidence of tumor-specific causality but rather as observations that may inform hypotheses for future, adequately powered studies. 43% of patients were diagnosed with sarcoidosis more than 5 years after being diagnosed with cancer. However, in our study, the duration of malignancy before sarcoidosis diagnosis was 4 years in EOS (median: 4.17 Y) and 3 years in younger-onset sarcoidosis (median: 3.3 Y), with no significant difference (*p* = 0.907). Mehta et al. [24] also reported 3 cases of breast cancer that had a simultaneous diagnosis of sarcoidosis. Two patients were diagnosed following the diagnosis of malignancy, and one had a pre-existing diagnosis. The patients belonged to later than usual ages (46, 55, and 65 years old (EOS)) and were asymptomatic for sarcoidosis. Fiorucci et al. [25] reported a case of sarcoidosis of the breast in a 51-year-old woman with systemic manifestations of sarcoidosis (arthralgias and uveitis) associated with a breast mass.

The present study demonstrates that although malignancy was more frequently observed among patients with elderly-onset sarcoidosis, this association was largely explained by age. When age was modeled as a continuous variable, it emerged as the sole independent predictor of prior malignancy, while EOS classification, smoking history, and comorbidity burden did not retain independent associations. These findings illustrate the importance of age-adjusted analyses when evaluating malignancy risk in EOS and suggest that the crude association between EOS and malignancy should not be interpreted as evidence of a distinct sarcoidosis subtype intrinsically linked to cancer.

EOS patients exhibited a higher prevalence of cardiometabolic comorbidities and smoking history, which may act as shared risk factors for both malignancy and sarcoidosis. However, after adjustment, these factors did not independently explain the association between malignancy and sarcoidosis timing, suggesting that their apparent influence is largely mediated through age. This finding highlights immune senescence and cumulative lifetime exposures as potential unifying mechanisms rather than individual comorbid conditions.

Despite the dominant effect of age, descriptive analyses revealed distinct tumor patterns in EOS patients, including higher frequencies of breast cancer in females and prostate cancer in males. These observations are consistent with age-dependent tumor epidemiology but may nonetheless carry clinical relevance in sarcoidosis populations.

Given the limited number of tumor-specific events, these findings should be interpreted cautiously and considered hypothesis-generating. Larger, age-matched studies are needed to determine whether sarcoidosis confers additional tumor-specific risk beyond that expected from aging alone.

Several biological mechanisms may underlie the observed coexistence of sarcoidosis and malignancy in elderly patients. Chronic inflammation and immune dysregulation associated with malignancy may promote granuloma formation, while immune senescence in older individuals may predispose them to both cancer development and dysregulated granulomatous responses [3].

In addition, cancer treatments such as chemotherapy, radiotherapy, and immunotherapy have been reported to induce sarcoid-like granulomatous reactions, potentially contributing to sarcoidosis onset in patients with prior malignancy. Although treatment-related effects were not formally modeled in the present study, this pathway represents an important area for future investigation [3].

Not only is there an association between malignancies and the development of sarcoidosis, but also the anticancer therapies can also induce granulomatous reactions, including sarcoidosis and sarcoid-like lesion (SLR) [18,26]. In a review of the WHO pharmacovigilance database, strong associations were found between SLR and several drugs, including pembrolizumab, nivolumab, and ipilimumab, as well as dabrafenib, vemurafenib, trametinib, and cobimetinib. These findings highlight the need for awareness and careful monitoring of granulomatous responses in patients undergoing these therapies [18,26]. In our study, patients with EOS receiving chemotherapy (5%), operative resection (23.3%), radiotherapy (5%), or adjuvant immunotherapy (2.5%) were significantly higher compared to younger-onset sarcoidosis (*p* = 0.035, ≤0.001, 0.013, 0.021). Some cancer treatments can induce granuloma formation.

Regarding the radiological presentation, 60.9% of EOS and 66.7% of younger-onset sarcoidosis patients with a history of malignancy presented with stage 1. Parenchymal infiltration was significantly higher among patients with no malignancy compared to cases with malignancy within both groups of younger and elderly-onset. In addition, 61.1% of patients with a history of malignancy were asymptomatic in younger-onset sarcoidosis, and 41.7% of them presented with dyspnea in EOS. Sarcoidosis was commonly detected at a younger stage in Arish et al. [23], presumably due to more systematic follow-up using computed tomography (CT) and positron emission tomography in cancer patients. Radiological findings were identical to those found in classic sarcoidosis (mediastinum and hilar lymphadenopathy). Most individuals had no symptoms when they were diagnosed with sarcoidosis. EOS in our study presented at later stages 3 and 4 compared to the younger-onset group (5.2%, 1.4%) (*p* ≤ 0.001). Rubio-Rivas et al. [14] also concluded that when compared to younger patients, older individuals with pulmonary sarcoidosis had more severe disease and poorer prognosis.

Clinically, dyspnea was more commonly observed in our study in EOS patients compared to the younger-onset group (*p* ≤ 0.001), while Loefgren syndrome was more common in the younger-onset group (*p* 0.005). The higher prevalence of dyspnea as the main symptom in EOS can be explained by the presence of comorbidities in this group of patients, such as COPD and CAD.

Finally, in our study, ex-smokers, hypertensive patients, COPD patients, and diabetic patients were more prevalent in the EOS group compared to the other group. There were significant differences between the two groups regarding the history of some other comorbidities, such as CAD and COPD. Kobak et al. [27] found that EOS patients had increased rates of fatigue comorbidities and fewer inflammatory signs.

It is interesting that the development of sarcoidosis following malignant disorders presents several challenges, particularly in establishing a definitive diagnosis, understanding the course of the disease, and predicting its prognosis. First, the clinical presentation of sarcoidosis frequently mimics the recurrence of malignancy, distant metastasis, or even side effects of various anticancer therapies. Furthermore, the routine use of CT and/or positron emission tomography (PET) in the follow-up of patients with a history of malignancy increases the detection of lymphadenopathy, which can resemble metastasis. Consequently, a histological diagnosis is crucial to establish a definitive diagnosis. Second, there is a belief that the development of secondary sarcoidosis may indicate a better prognosis for the malignancy, as it is thought to represent a strong immune response to the tumor cells. This type of reaction has been primarily observed in lymphoma and testicular cancer [18]. However, these observations remain controversial, and further studies are needed, particularly in relation to other types of malignancy. Third, sarcoidosis remains a highly heterogeneous disease, with variability in clinical presentation, natural history, and response to treatment. Its course can range from spontaneous remission in up to two-thirds of patients to progression into chronic disease with significant morbidity and permanent disability [28]. Due to this heterogeneity, stratifying patients into clusters or phenotypes, such as idiopathic versus secondary sarcoidosis, may be useful in predicting the clinical course and outcome.

The primary limitations of this study are its retrospective design and single-center setting, which inherently restrict generalizability and are further compounded by the relatively small sample size. The number of malignancy events restricted the complexity of multivariable modeling. Absence of interval-timing analysis is another limitation. Additionally, tumor-specific and treatment-related analyses were limited by sample size. Another important limitation is the inability to determine whether the observed differences in tumor history are genuinely related to the EOS phenotype or are instead a consequence of age disparities between groups. This challenge is amplified by the lack of an age-matched control cohort without EOS, which limits the robustness of any causal interpretation. To overcome these constraints, future investigations should employ age-matched case–control or prospective cohort designs that allow more rigorous adjustment for potential confounders and facilitate a clearer understanding of the relationship between EOS subtypes and tumor history.

## 5. Conclusions

In conclusion, elderly-onset sarcoidosis (EOS) is a less frequent variant of sarcoidosis with limited data regarding the possible risk factors. The increased prevalence of malignancy observed among patients with elderly-onset sarcoidosis appeared to be largely driven by age rather than a distinct EOS-specific effect. Age-adjusted analyses are essential when interpreting malignancy risk in sarcoidosis, and future age-matched prospective studies are needed to clarify potential biological links and guide evidence-based screening strategies.

From a practical standpoint, our findings highlight the need for heightened clinical awareness of malignancy in elderly patients at the time of sarcoidosis diagnosis. Although routine malignancy screening cannot be recommended based on the present data, future prospective studies should evaluate whether targeted cancer assessment at EOS diagnosis may be clinically beneficial. Given the tumor distribution observed in this cohort, particular attention in such studies may be directed toward breast and ovarian cancer in women and prostate cancer in men. Non-invasive diagnostic modalities, including breast ultrasound, mammography, prostate-specific antigen testing, and abdominal imaging, may represent feasible tools for evaluation; however, their clinical utility and cost-effectiveness in EOS populations require prospective validation.

## Figures and Tables

**Figure 1 arm-94-00003-f001:**
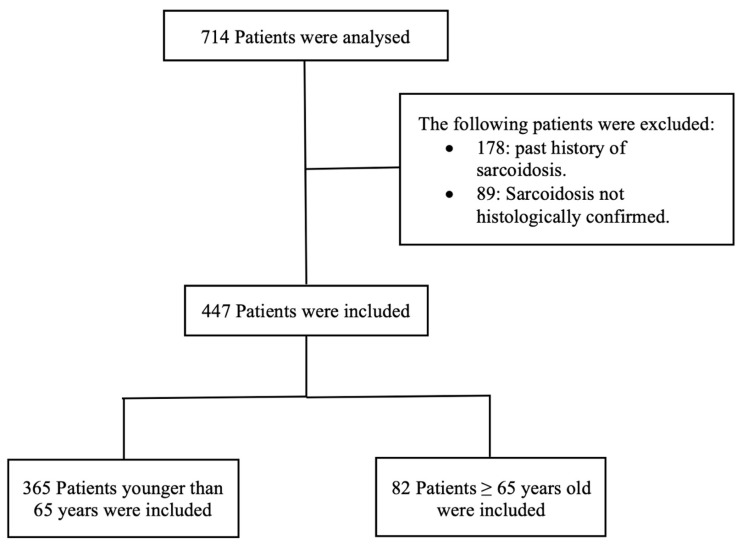
Flowchart of the study.

**Figure 2 arm-94-00003-f002:**
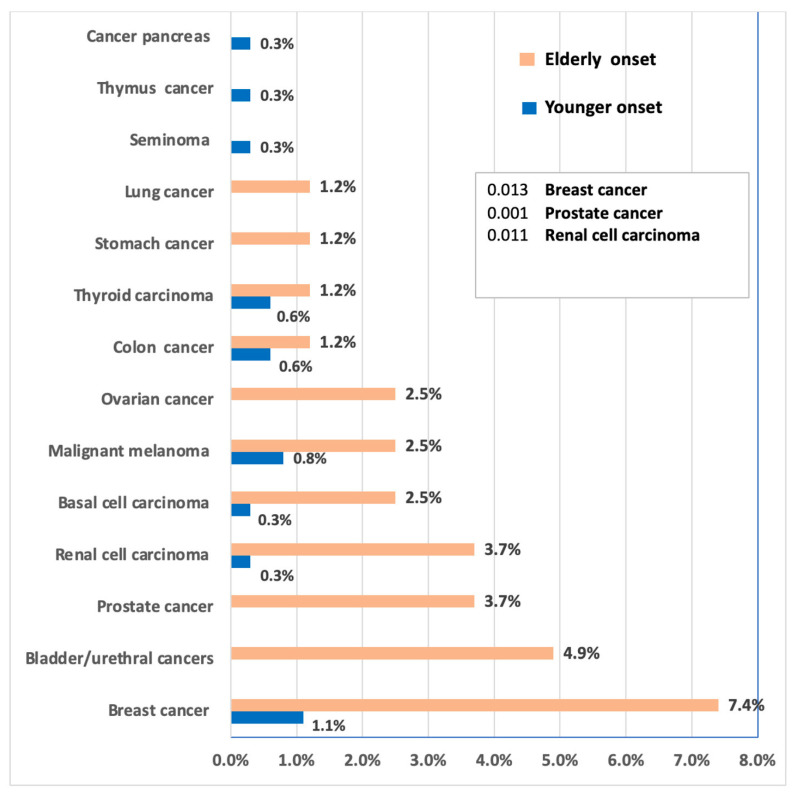
Types of malignancy in association with the onset of sarcoidosis. NB: The percentages shown in the figure represent the percentage of the tumor in the total groups of younger and elderly-onset sarcoidosis. *p* values are the significant differences between the two groups. Other differences were not significant. For breast cancer, analysis was performed only including females, while for prostate cancer, analysis was performed including males only (among females only, breast cancer was present in 15.4% in the elderly-onset group and 3.3% in the younger-onset group, while ovarian cancer was present in 5.7% in the elderly-onset group). For prostate cancer, it represented 11.1% of the elderly-onset group males.

**Table 1 arm-94-00003-t001:** The clinical characteristics of the studied patients in association with the onset of sarcoidosis.

	Younger-Onset N = 365		Elderly-Onset N = 82		Crude OR (95% CI)	*p* Value	AOR (95% CI)	*p* Value
	N	%	N	%				
**Age: median [IQR]**	47 [23–63]		69 [65–84]			*p* ≤ 0.001	___	____
**Sex**					r (1)		r (1)	
Female	131	35.9%	45	54.9%	2.2 (1.3–3.5)	0.002	2.1 (1.0–4.3)	0.041
**Smoking status**								
Non-smoker	148	53%	25	45.5%	r (1)		r (1)	
Ex-smoker	66	23.7%	27	49.1%	2.4 (1.3–4.4)	0.005	1.8 (0.9–3.9)	0.117
Current smoker	65	23.3%	3	5.5%	0.3 (0.1–0.9)	0.039	0.2 (0.06–0.9)	0.044
**HTN**	72	19.8%	59	75.6%	12.6 (7.0–2.4)	≤0.001	8.1 (3.9–16.8)	≤0.001
**DM**	30	8.3%	23	30.3%	4.8 (2.6–8.9)	≤0.001	3.3 (1.3–8.7)	0.013
**Coronary artery disease**	6	1.6%	8	9.8%	6.5 (2.2–19.2)	0.001	3.9 (0.7–21.7)	0.121
**Hypothyroidism**	11	3.0%	6	7.3%	2.5 (0.9–7.1)	0.075	___	
**OSA**	10	2.7%	0	0.0%	-	0.220	___	
**COPD**	6	1.6%	6	7.3%	4.7 (1.5–15.0)	0.009	6.0 (1.1–32.0)	0.038
**Bronchial asthma (BA)**	28	7.7%	2	2.4%	0.3 (0.1–1.3)	0.106	___	

**Table 2 arm-94-00003-t002:** The clinical presentation, radiological signs in the CT chest, and functional parameters of the sarcoidosis in association with the onset of the disease.

	Younger-Onset N = 365		Elderly-Onset N = 82		*p* Value
	N	%	N	%	
**Leading symptoms**					
Accidental	80	22.2%	20	24.7%	0.623
Cough	137	38.0%	23	28.4%	0.106
Dyspnea	70	19.4%	41	50.6%	0.001
Chest pain	45	12.5%	2	2.5%	0.08
Loefgren syndrome	49	13.6%	2	2.5%	0.005
Weight loss	10	2.8%	3	3.7%	0.714
Conjunctivitis/iritis/iridocyclitis/optic neuritis/uveitis	8	2.2%	0	0.0%	0.360
Night sweat	2	0.6%	1	1.2%	*p*: 0.456 ^a^
AV block	1	0.3%	0	0.0%	*p*: 1 ^a^
Tachycardia	1	0.3%	0	0.0%	*p*: 1 ^a^
Fever	3	0.8%	1	1.2%	*p*: 0.556 ^a^
Hemoptysis	3	0.8%	0	0.0%	*p*: 1 ^a^
Horseness of voice (HOV)	1	0.3%	0	0.0%	*p*: 1 ^a^
Herfordt syndrome	1	0.3%	0	0.0%	*p*: 1 ^a^
**CT chest morphological signs**					
Enlarged LN	343	94.5%	73	89.0%	0.070
Parenchymal infiltration	221	61.0%	46	56.1%	0.408
Radiological staging according to Scadding					
○ 0	0	0.0%	2	2.5%	≤0.001 *
○ I	146	40.2%	37	45.7%	
○ II	193	53.2%	27	33.3%	
○ III	19	5.2%	10	12.3%	
○ IV	5	1.4%	5	6.2%	
**Lung function parameters**					
FEV1 (L) Mean (SD)	3.19 (0.93)		2.27 (0.85)		≤0.001 ^b^
FEV1 (%) Median (min–max)	90 (4–138)		87 (27–139)		0.270 ^c^
FVC (L) Median (min–max)	3.8 (0.37–89)		2.7 (0.9–4.9)		≤0.001 ^c^
FVC (%) Median (min–max)	90 (12–139)		84 (1.9–123)		0.117 ^c^
DLCO Median (%) (min–max)	77 (26–132)		61 (10–108)		≤0.001 ^c^

^a^ Fisher’s exact test, ^b^ Welch’s t-test, ^c^ Mann–Whitney U test; otherwise, a chi-square test was used. * Post hoc comparisons showed that compared to stage 0, there were significant differences in stages (2) and (3) (*p* = 0.044, 0.017, respectively), while there were no significant differences in stages (3) and (4) (*p* = 0.142, 0.470).

**Table 3 arm-94-00003-t003:** The diagnostic method of sarcoidosis and the indication for treatment after multidisciplinary discussion (MDD) in association with the onset of the disease.

	Younger-Onset N = 365		Elderly-Onset N = 82		*p* Value
	N	%	N	%	
**Diagnostic methods**					
Bronchoscopic procedures:					
EBUS-TBNA	265	72.6%	50	61%	0.037
EBB	44	12.1%	2	2.4%	0.010
BAL	71	19.5%	1	1.2%	0.001
TBB	147	40.3%	21	25.6%	0.013
Mediastinoscopy	34	9.3%	12	14.6%	0.152
VATs/open lung biopsy	4	1.1%	4	4.9%	0.041 ^a^
CT-guided biopsy	3	0.8%	1	1.3%	0.557 ^a^
LN resection	1	0.3%	2	2.4%	0.088 ^a^
**Treatment after multidisciplinary discussion (MDD)**	127	35.1%	32	39.5%	0.453

^a^ Fisher’s exact test was used; otherwise, a chi-squared test was used.

**Table 4 arm-94-00003-t004:** The association of sarcoidosis with a previous history of malignancy.

	Younger-Onset N = 365		Elderly-Onset N = 82		Crude OR (95% CI)	*p* Value
	N	%	N	%		
**History of malignancy**						
No history	345	95.0%	57	70.4%	r (1)	
Positive history	18	5.0%	24	29.6%	8.1 (4.1–15.8)	≤0.001
**Number of malignancies**						
1	18	5.0%	17	21%	5.7 (2.8–11.7)	≤0.001
2	0	0.0%	6	7.4%	-	≤0.001
3	0	0.0%	1	1.2%	-	0.144
**Treatment of malignancy**						
Chemotherapy	6	1.7%	4	5.0%	4.0 (1.1–14.7)	0.035
Operative resection	14	3.9%	19	23.8%	8.2 (3.9–17.3)	≤0.001
Radiotherapy	4	1.1%	4	5.0%	6.0 (1.4–24.9)	0.013
Adjuvant immunotherapy	0	0.0%	2	2.5%	-	0.021
**Duration of malignancy before Y median (min**–**max)**	3.3 (0–26)		4.17 (0.08–20)		1.0 (0.9–1.1)	0.907

NB: All items of the history of malignancy are compared with the reference of (no history of malignancy).

**Table 5 arm-94-00003-t005:** Sex-stratified distribution of malignancy types according to sarcoidosis onset group.

Tumor Type	<65 n (%)	≥65 n (%)	OR (95% CI)	*p* Value
**For females (N = 174)**				
No malignancy	119 (91.5)	33 (75.0)	Reference	—
Breast cancer	4 (3.1)	6 (13.6)	5.41 (1.43–20.4)	0.011
Ovarian cancer	0 (0.0)	2 (4.5)	9.67 (0.45–208.3)	0.048
Other tumors	7 (5.4)	3 (6.8)	1.55 (0.37–6.50)	0.54
**For males (N = 270)**				
**No malignancy**	226 (97.0)	24 (64.9)	Reference	—
**Prostate cancer**	0 (0.0)	3 (8.1)	**29.3 (1.45–593)**	0.006
Other tumors	7 (3.0)	10 (27.0)	**13.4 (4.7–38.1)**	<0.001
**For all patients**				
**No malignancy**	345 (95.0)	57 (70.4)	Reference	—
Renal cell carcinoma	1 (0.3)	3 (3.7)	18.2 (1.85–179)	0.013
Bladder/urethral	0 (0.0)	4 (4.9)	55.6 (2.9–1056)	0.001
Malignant melanoma	3 (0.8)	2 (2.5)	4.03 (0.67–24.1)	0.12
Colon cancer	2 (0.6)	1 (1.2)	3.03 (0.27–33.6)	0.36
Lung cancer	0 (0.0)	1 (1.2)	14.5 (0.57–368)	0.09

Fisher’s exact test.

**Table 6 arm-94-00003-t006:** Differences between the patients with positive and negative histories of malignancy within each of the groups of the study (younger and elderly-onset).

	Younger-Onset N = 365		*p* Value	Elderly-Onset N = 82		*p* Value
	Negative Malignancy N (%)	Positive Malignancy N (%)		Negative Malignancy N (%)	Positive Malignancy N (%)	
**Age:** median (min–max)	47 (23–63)	53.5 (35–61)	0.010	69 (65–84)	67.5 (65–80)	0.427
**Sex**						
Female	119 (34.5)	11 (61.1)	0.022	33 (57.9)	11 (45.8)	0.320
**Smoking:**						
Non-smoker	138 (52.1)	10 (76.9)	0.240	17 (43.6)	8 (50)	0.892
Ex-smoker	64 (24.2)	1 (7.7)		20 (51.3)	7 (43.8)	
Current	63 (23.8)	2 (15.4)		2 (5.1)	1 (6.3)	
**HTN**	67 (19.5)	4 (22.2)	0.762	40 (75.5)	18 (75)	0.965
**DM**	29 (8.5)	1 (5.6)	1	15 (28.8)	8 (34.8)	0.607
**CAD**	5 (1.4)	1 (5.6)	0.265	3 (5.3)	5 (20.8)	0.046
**Hypothyroidism**	10 (2.9)	1 (5.6)	0.433	4 (7.0)	2 (8.3)	1
**OSA**	10 (2.9)	0 (0)	1	0	0	-
**COPD**	6 (1.7)	0 (0)	1	5 (8.8)	1 (4.2)	0.664
**BA**	27 (7.8)	1 (5.6)	1	2 (3.5)	0 (0)	1
**Leading symptoms**						
Accidental	68 (19.9)	11 (61.1)	0	11 (19.6)	9 (37.5)	0.091
Cough	135 (39.5)	2 (11.1)	0.016	17 (30.4)	5 (20.8)	0.382
Dyspnea	70 (20.5)	0 (0)	0.030	30 (53.6)	10 (41.7)	0.329
Chest pain	43 (12.6)	2 (11.1)	1	1 (1.8)	1 (4.2)	0.513
Löfgren syndrome	47 (13.7)	2 (11.1)	1	1 (1.8)	1 (4.2)	0.513
Weight loss	9 (2.6)	1 (5.6)	0.405	2 (3.6)	1 (4.2)	1
Conjunctivitis/iritis/iridocyclitis/optic neuritis/uveitis	8 (2.3)	0	0.967	-	-	
Night sweat	2 (0.6)	0	0.405	1 (1.8)	0	0.510
AV block	1 (0.3)	0	0.270	0	0	
Tachycardia	1 (0.3)	0	0.270	0	0	
Fever	3 (0.9)	0	0.528	0	0	
Hemoptysis	3 (0.9)	0	0.528	0	0	
HOV	1 (0.3)	0	0.270	0	0	
Herfordt syndrome	1 (0.3)	0	0.270	0	0	
**Diagnostic methods**						
EBUS-TBNA	254 (73.6)	11 (61.1)	0.277	32 (56.1)	18 (75)	0.111
EBB	43 (12.5)	1 (5.6)	0.709	2 (3.5)	0	1
BAL	69 (20)	2 (11.1)	0.544	1 (1.8)	0	1
TBB	141 (40.9)	5 (27.8)	0.269	18 (31.6)	3 (12.5)	0.074
Mediastinoscopy	30 (8.7)	4 (22.2)	0.076	10 (17.5)	1 (4.2)	0.160
VATs/open lung biopsy	2 (0.6)	1 (5.6)	0.142	2 (3.5)	2 (8.3)	0.578
CT-guided biopsy	2 (0.6)	1 (5.6)	0.142	1 (1.8)	0	1
LN resection	1 (0.3)	0	1	1 (1.8)	1 (4.2)	0.507
**Lung function parameters**						
FEV1L Median (min–max)	3.2 (0.4–5.8)	3 (1.4–5.1)	0.318	2.2 (0.7–4.3)	2.4 (1.1–4.5)	0.169
FEV1% Median (min–max)	90 (4.2–138)	96.5 (48–134)	0.080	86 (27–139)	90.5 (47–138)	0.422
FVCL Median (min–max)	3.8 (0.4–89)	3.5 (1.6–6)	0.117	2.6 (0.9–4.9)	3 (1.3–4.7)	0.306
FVC % Median (min–max)	90 (12–139)	89.5 (45–135)	0.392	82.5 (29–123)	84.5 (1.9–118)	0.513
DLCO Median (min–max)	77 (26–132)	82 (40–92)	0.827	58.5 (10–108)	74 (18–95)	0.324
**CT chest morphological signs**						
Enlarged LN	326 (94.6)	17 (94.4)	1	51 (89.5)	21 (87.5)	1
Parenchymal infiltration	214 (62.4)	6 (33.3)	0.014	36 (63.2)	9 (37.5)	0.034
Radiological staging according to Scadding						
○ 0	0	0	0.102	1 (1.8)	1 (4.3)	0.254
○ I	134 (39)	12 (66.7)		23 (40.4)	14 (60.9)	
○ II	187 (54.4)	5 (27.8)		21 (36.8)	5 (21.7)	
○ III	18 (5.2)	1 (5.6)		7 (12.3)	3 (13)	
○ IV	5 (1.5)	0 (0)		5 (8.8)	0 (0)	
**Treatment after MDD**	125 (36.4)	2 (11.1)	0.028	23 (41.1)	8 (33.3)	0.515

**Table 7 arm-94-00003-t007:** Multivariable logistic regression analysis of factors associated with prior malignancy in sarcoidosis patients.

Variable	Adjusted OR	95% CI	*p* Value
Age (per 1-year increase)	1.08	1.04–1.12	<0.001
Male sex	1.33	0.58–3.04	0.50
Diabetes and/or hypertension	1.55	0.61–3.89	0.36
Smoking (ever vs. never)	0.58	0.25–1.36	0.21

## Data Availability

The data generated in this study are available upon request from the corresponding author.

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
