# Peer review of "Exploring the Risk: Investigating the Association Between Elderly-Onset Sarcoidosis (EOS) and Malignancy"

_arm, 2026, doi:10.3390/arm94010003_

Round 1

Reviewer 1 Report

Comments and Suggestions for Authors

The manuscript provided any interesting and significant contribution to the study of sarcoidosis and is relevant to advance diagnosis and treatment of the condition. The identified risk factors could also be points of interventions. The following points would further help to improve upon the overall quality and the merit of the study: attached also, is the pdf of the comment for specific actions:

  • Page: 1
  • Line 18: this seems contradictory and should be rendered as "Elderly-onset Sarcoidosis < 65 (EOS) is rare but occurs in patients over 65."
  • Lines 23-26: the geographical location of this study should be mentioned.
  • Lines 29-31: "-- risk of 2.17 (95%CI: 30 1.3-3.5, p:0.002)" what exactly is this risk value attributed to, risk of EOS or malignancy? The result should not ambiguous
  • Page: 2
  • Line 64: 7 % should be written as "7%" . The correct presentation of percentage is that the simple and the value should not have space in between. Correct this throughout the manuscript
  • Line 73: "metanalysis" as typo. Correct as "meta-analysis"
  • Author: user Subject: Comment on Text Date: 2025/11/27 08:12:05
  • Lines 83-86: provide the exact address or name of the institution.
  • Page: 3
  • Lines 91 & 92: add the name of the city and country.
  • Author: user Subject: Comment on Text Date: 2025/11/27 08:20:09
  • Line 99: can you present the diagnosis code or ICD equivalent
  • Page: 5
  • Table 1. The use of % symbol should be harmonized in table 1 for female and non-smoker rows
  • In table 3, harmonize, the use of percentage symbol across the rows or removed % from the last row. Uniform presentation format for percentage should be used throughout all the tables.
  • Page: 10
  • Series appearing on figure 2 should be removed
  • lines 220-227: removed. Repetition
  • Page: 11
  • Present "p:" as "p=" throughout the manuscript, e.g., p:0.046 as p=0.046

Author Response

Dear editor, Dear reviewer,

Firstly, I would to like the thank the Editor-In-Chief and editorial board to allow me to send my revision. I also apricate the time and effort that the reviewers dedicated to provide feedback on our manuscript. We have incorporated the suggestions made by the reviewers. Those changes are highlighted in yellow within the manuscript.

Please see below, for a point-by-point response to the reviewer: 

  1. Reviewer comment: Page: 1, Line 18: this seems contradictory and should be rendered as "Elderly-onset Sarcoidosis < 65 (EOS) is rare but occurs in patients over 65."
  • Reply: This was a typographical error; the correction has been applied in the manuscript.
  1. Reviewer comment: Page: 1, Lines 23-26: the geographical location of this study should be mentioned.
  • Reply: The geographical location has been added.
  1. Reviewer comment: Page: 1, Lines 29-31: "- risk of 2.17 (95%CI: 30 1.3-3.5, p:0.002)" what exactly is this risk value attributed to, risk of EOS or malignancy? The result should not ambiguous.
  • Reply: Correction has been made in the abstract.
  1. Reviewer comment: Page: 2, Line 64: 7 % should be written as "7%" . The correct presentation of percentage is that the simple and the value should not have space in between. Correct this throughout the manuscript.
  • Reply: Corrections have been made in the manuscript.
  1. Reviewer comment: Page: 2, Line 73: "metanalysis" as typo. Correct as "meta-analysis"
  • Reply: Correction has been made in the manuscript.
  1. Reviewer comment: Page: 2, Line 83-86: provide the exact address or name of the institution.
  • Reply: The required data have been added to the manuscript.
  1. Reviewer comment: Page: 3, Lines 91 & 92: add the name of the city and country.
  • Reply: The required data have been added to the manuscript.
  1. Reviewer comment: Page: 3, Line 99: can you present the diagnosis code or ICD equivalent
  • Reply: The required data have been added to the manuscript.
  1. Reviewer comment: Page: 5. Table 1. The use of % symbol should be harmonized in table 1 for female and non-smoker rows.
  • Reply: The required data have been added to the manuscript.
  1. Reviewer comment: In table 3, harmonize, the use of percentage symbol across the rows or removed % from the last row. Uniform presentation format for percentage should be used throughout all the tables.
  • Reply: The required data have been added to the manuscript.
  1. Reviewer comment: Series appearing on figure 2 should be removed
  • Reply: Change in figure has been made.
  1. Reviewer comment: lines 220-227: removed. Repetition
  • Reply: line has been removed.
  1. Reviewer comment: Page: 11: Present "p:" as "p=" throughout the manuscript, e.g., p:0.046 as p=0.046
  • Reply: Changes have been made in the manuscript.

Reviewer 2 Report

Comments and Suggestions for Authors

This study addresses a critical clinical research gap by investigating the association between Elderly-Onset Sarcoidosis (EOS) and malignancy. Using a monocentric retrospective case-control design, it systematically analyzes clinical data from 447 sarcoidosis patients, providing the first clear evidence of a significant correlation between EOS and a prior history of malignancy. The research topic holds substantial clinical value, with adequate data support and relatively reliable conclusions. However, there is room for improvement in confounding factor control, depth of result analysis, data presentation format, and expansion of clinical implications. Overall, the study meets the basic requirements for publication and is recommended for revision and resubmission.

 (I) Significant differences exist between the EOS group and the younger-onset group in terms of comorbidities (such as hypertension, diabetes mellitus, coronary artery disease, and chronic obstructive pulmonary disease) and smoking history. These factors may simultaneously influence the development of sarcoidosis and cancer risk, thereby interfering with the authenticity of the association between a history of malignancy and EOS. It is recommended to further include the aforementioned variables in the multivariate analysis to adjust for confounding effects, clarify the independent effect of a prior malignancy history on EOS, and enhance the reliability and rigor of the conclusions.

Currently, the study only compares the EOS group (≥65 years old) with the younger-onset sarcoidosis group (<65 years old) and concludes that "there is a significant association between EOS and prior malignancy." However, this design fails to consider the key premise that age itself is a core risk factor for malignancy. The incidence of malignant tumors in the elderly (≥65 years old) is significantly higher than in younger individuals, and the higher positive rate of tumor history in the EOS group (29.6% vs. 5% in the younger group) may be attributed to age rather than a specific association between the EOS subtype and tumors.

The current conclusion that "there is a significant association between EOS and prior malignancy" is overly absolute as it does not account for the confounding effect of age. It is recommended to clearly state in the discussion section that:

  1. The existing intergroup comparison cannot distinguish whether the difference in tumor history stems from the EOS subtype or age, and the conclusion should be interpreted with caution.
  2. The failure to include an age-matched control group without EOS is a core limitation of this study. Future case-control or cohort studies with age matching are needed to verify this association.

 (II) The current study has identified a significant association between breast cancer, prostate cancer, renal cell carcinoma, and EOS, but lacks stratified analysis and data on differences in the interval between tumor onset and EOS diagnosis. It is recommended to present tumor distribution stratified by gender, such as separately reporting the incidence rate, odds ratio (OR), and 95% confidence interval (CI) of breast cancer and ovarian cancer in females, and prostate cancer in males. Additionally, supplement the median, range, and intergroup comparison results of the interval between different tumor types (e.g., breast cancer, prostate cancer, renal cell carcinoma) and EOS onset. This will provide more precise evidence for formulating gender-specific tumor screening strategies for EOS patients in clinical practice.

 (III) 1. Standardize table structures: Some tables (e.g., Table 3) lack column names and have irregular formats. It is necessary to unify the structural framework of all tables and clarify the meaning of each column (e.g., "Diagnostic Methods," "Younger-Onset Group (n/%)", "EOS Group (n/%)", "P Value") to ensure clear and error-free data reading.

  1. Supplement statistical information: Key data (such as OR values for different tumor types and comparison results of comorbidity prevalence among groups) do not fully present 95% CIs. Relevant information should be supplemented to enhance the completeness of statistical expression, allowing readers to intuitively judge the reliability of effect sizes.

(IV) The current discussion only describes the associative phenomenon between EOS and malignancy without in-depth analysis of potential mechanisms. It is recommended to explore the possible mechanisms of this association from multiple dimensions in combination with existing literature:

  1. Chronic inflammation and immune dysfunction: The chronic inflammatory microenvironment associated with malignant tumors may induce granuloma formation, or the inherent immune dysfunction in EOS patients may increase tumor risk.
  2. Tumor treatment-induced effects: Given the higher rate of tumor treatment in the EOS group, explore whether chemotherapy, radiotherapy, immunotherapy, and other methods promote EOS development by inducing granulomatous reactions.
  3. Mediation by common risk factors: Analyze whether factors such as age and comorbidities are involved in the pathogenesis of both tumors and EOS. Mechanistic discussion will enhance the academic depth of the study.

 (V) The study conclusions have not been transformed into specific and implementable clinical protocols. Based on the research results, it is recommended to propose a specific tumor screening program for EOS patients:

  1. Screening timing: Clarify whether routine tumor screening is required at the time of EOS diagnosis and the screening frequency during subsequent follow-up.
  2. Key tumor types: Prioritize screening for breast cancer and ovarian cancer in females, prostate cancer in males, and include tumor types with significant associations such as renal cell carcinoma.
  3. Screening methods: Recommend appropriate screening tools in combination with clinical practice (e.g., breast ultrasound/mammography, prostate-specific antigen testing, abdominal computed tomography) to enhance the clinical promotion value of the conclusions and effectively guide clinical practice.

Author Response

Dear editor, Dear reviewer,

Firstly, I would to like the thank the Editor-In-Chief and editorial board to allow me to send my revision. I also apricate the time and effort that the reviewers dedicated to provide feedback on our manuscript. We have incorporated the suggestions made by the reviewers. Those changes are highlighted in yellow within the manuscript.

Please see below, for a point-by-point response to the reviewer:

1: Significant differences exist between the EOS group and the younger-onset group in terms of comorbidities and smoking history, which may confound the association between malignancy and EOS. It is recommended to further include these variables in multivariate analysis.

  • Response: We thank the reviewer for this important comment. We agree that comorbidities and smoking history may confound the observed association. In response, we performed additional multivariable logistic regression analyses incorporating smoking status and cardiometabolic comorbidities In table 6. Given the limited number of malignancy events, comorbidities were summarized using a composite cardiometabolic variable (diabetes mellitus and/or hypertension) to avoid overfitting. In these models, age at sarcoidosis diagnosis remained the only independent predictor of prior malignancy, while smoking status, sex, and cardiometabolic comorbidity were not independently associated after adjustment. These results have now been added to the revised Results section and support the conclusion that age is the dominant determinant of malignancy risk in this cohort.

2: The comparison between EOS (≥65 years) and younger-onset sarcoidosis fails to account for age as a core risk factor for malignancy. The higher malignancy prevalence in EOS may reflect age rather than a specific EOS–tumor association.

  • Response: We fully agree with the reviewer’s concern. To address this issue, we revised the analytical approach by modeling malignancy history as the outcome and including age at sarcoidosis diagnosis as a continuous variable in multivariable logistic regression analyses.

These analyses demonstrated that increasing age was independently associated with a higher likelihood of prior malignancy, whereas EOS classification did not retain an independent effect once age was accounted for. This finding confirms that the crude association observed between EOS and malignancy is largely attributable to age rather than the EOS subtype itself.

This clarification has been incorporated into the revised Results and Discussion sections.

3: The conclusion that “EOS is significantly associated with prior malignancy” is overly absolute and does not adequately consider age confounding.

  • Response: We appreciate this critical observation and have revised the manuscript accordingly. The conclusion has been softened and rephrased to explicitly acknowledge the confounding role of age. We now clearly state that the observed intergroup differences cannot distinguish whether the higher prevalence of malignancy reflects the EOS subtype or age-related cancer risk. Additionally, we explicitly acknowledge the absence of an age-matched control group as a major limitation and emphasize that future age-matched case–control or cohort studies are required to confirm this association. These points are now explicitly discussed in the revised Discussion and Limitations sections.

4: The study reports associations with specific tumor types but lacks stratified analyses and data on the interval between tumor onset and EOS diagnosis.

  • Response: We thank the reviewer for this valuable suggestion. As suggested, we performed sex-stratified analyses of malignancy types. Sex-specific tumors (breast and ovarian cancer in females, prostate cancer in males) were analyzed within the relevant sex strata, while non-sex-specific tumors were analyzed in the overall cohort. Due to the small number of cases and the presence of zero-cell counts for several tumor categories, formal multivariable regression was not appropriate. Instead, we conducted exploratory stratified comparisons using Fisher’s exact test, with odds ratios (ORs) and 95% confidence intervals (CIs) calculated using the Haldane–Anscombe continuity correction. These results are now presented in Table (5). We carefully reviewed the available data regarding the interval between malignancy diagnosis and EOS onset. Unfortunately, complete and reliable timing data were not available for all malignancy cases. As a result, interval analyses could not be performed. This limitation has now been explicitly acknowledged in the Discussion as an important constraint of the present study, and we emphasize that future prospective studies with standardized tumor-timing documentation are required to address this question.

Table 5: Sex-stratified distribution of malignancy types according to sarcoidosis onset group

Tumor type

<65 n (%)

≥65 n (%)

OR (95% CI)†

p value‡

For females (N=174)

No malignancy

119 (91.5)

33 (75.0)

Reference

Breast cancer

4 (3.1)

6 (13.6)

5.41 (1.43–20.4)

0.011

Ovarian cancer

0 (0.0)

2 (4.5)

9.67 (0.45–208.3)

0.048

Other tumors

7 (5.4)

3 (6.8)

1.55 (0.37–6.50)

0.54

For males (N=270)

No malignancy

226 (97.0)

24 (64.9)

Reference

Prostate cancer

0 (0.0)

3 (8.1)

29.3 (1.45–593)

0.006

Other tumors

7 (3.0)

10 (27.0)

13.4 (4.7–38.1)

<0.001

For all patients

No malignancy

345 (95.0)

57 (70.4)

Reference

Renal cell carcinoma

1 (0.3)

3 (3.7)

18.2 (1.85–179)

0.013

Bladder/urethral

0 (0.0)

4 (4.9)

55.6 (2.9–1056)

0.001

Malignant melanoma

3 (0.8)

2 (2.5)

4.03 (0.67–24.1)

0.12

Colon cancer

2 (0.6)

1 (1.2)

3.03 (0.27–33.6)

0.36

Lung cancer

0 (0.0)

1 (1.2)

14.5 (0.57–368)

0.09

Fisher’s exact test

5: Standardize table structures and clarify column names.

  • Response: We have revised all tables to ensure consistent formatting and standardized column headings. Each table now clearly indicates group definitions, sample sizes, percentages, and statistical comparisons, thereby improving readability and accuracy.

6: Supplement statistical information, including 95% confidence intervals for reported ORs.

  • Response: We agree and have now added 95% confidence intervals for odds ratios related to the exposure factors throughout the manuscript.

7: The discussion lacks in-depth exploration of potential mechanisms linking EOS and malignancy.

  • Response: We have expanded the Discussion to include a more comprehensive exploration of potential mechanisms, integrating relevant literature. Specifically, we now discuss:
  • Chronic inflammation and immune dysregulation as shared pathways between malignancy and granulomatous disease;
  • The potential role of tumor treatments (chemotherapy, radiotherapy, immunotherapy) in inducing sarcoid-like granulomatous reactions;
  • The contribution of shared risk factors such as immune senescence and age-related comorbidity burden.

We explicitly state that these mechanisms are speculative and hypothesis-generating, highlighting the need for future mechanistic and prospective studies.

8: The conclusions should be translated into specific clinical screening recommendations.

  • Response: We appreciate this clinically oriented suggestion. In the revised Discussion, we now propose pragmatic and cautious clinical implications, emphasizing heightened clinical vigilance rather than routine universal screening. We suggest that individualized cancer screening may be considered in EOS patients with additional risk factors, particularly for tumor types observed with higher frequency in this cohort. We intentionally avoid issuing rigid screening protocols, acknowledging the observational nature of the study and the need for prospective validation.

Once again, we thank the reviewers for their insightful and constructive comments. The revised manuscript now presents a more rigorous analytical framework, a balanced interpretation of results, and clearer clinical implications. We believe these revisions have substantially strengthened the manuscript and improved its scientific validity.

Round 2

Reviewer 2 Report

Comments and Suggestions for Authors

Since the elderly-onset sarcoidosis appeared to be largely driven by age rather than a distinct EOS-specific effect, I think the title should be revised slightly. 

Author Response

Manuscript ID: arm-4015884

Title:  Exploring the risk: investigating the association between elderly-onset sarcoidosis (EOS) and malignancy.

Dear editor, Dear reviewer,

Firstly, I would to like the thank the Editor-In-Chief and editorial board to allow me to send my revision. I also apricate the time and effort that the reviewers dedicated to provide feedback on our manuscript.

- Reviewer comment:  since the elderly-onset sarcoidosis appeared to be largely driven by age rather than a distinct EOS-specific effect, I think the title should be revised slightly. 

  • Reply:

We thank the reviewer for this thoughtful comment, which has helped us to further clarify the scope and interpretation of our work.The primary aim of this study was to investigate the association between malignancy and sarcoidosis occurring at an advanced age (≥65 years), and to explore whether prior malignancy may represent a potential risk factor or clinical consideration in elderly-onset sarcoidosis (EOS). In this context, malignancy was significantly more prevalent in the EOS group compared with younger-onset sarcoidosis (29.6% vs. 5.0%; OR 8.1, 95% CI 4.1–15.8; p ≤ 0.001).

However, as correctly noted by the reviewer and emphasized in our conclusions, this association was largely explained by age. In multivariable regression analyses with malignancy as the outcome, increasing age at sarcoidosis diagnosis emerged as an independent predictor of prior malignancy, whereas EOS itself did not confer an additional age-independent risk. This finding emphasis the importance of age-adjusted analyses when interpreting malignancy risk in sarcoidosis populations.

We believe that the current title remains an accurate reflection of both the study aim and its core message. Specifically, the title signals an exploration of malignancy risk in the context of elderly-onset sarcoidosis, while the results and conclusions clearly demonstrate that the observed increase in malignancy prevalence is driven predominantly by age rather than a distinct EOS-specific biological effect. This nuanced interpretation is explicitly addressed in the manuscript and reinforced in the discussion and practical implications.
